# Metformin and Its Immune-Mediated Effects in Various Diseases

**DOI:** 10.3390/ijms24010755

**Published:** 2023-01-01

**Authors:** Ichiro Nojima, Jun Wada

**Affiliations:** Department of Nephrology, Rheumatology, Endocrinology and Metabolism, Faculty of Medicine, Dentistry and Pharmaceutical Sciences, Okayama University, Okayama 700-8558, Japan

**Keywords:** CD8 T cells, AMPK, mTORC, OXPHOS, autoimmune disease, aging, cancer

## Abstract

Metformin has been a long-standing prescribed drug for treatment of type 2 diabetes (T2D) and its beneficial effects on virus infection, autoimmune diseases, aging and cancers are also recognized. Metformin modulates the differentiation and activation of various immune-mediated cells such as CD4+ and CD+8 T cells. The activation of adenosine 5′-monophosphate-activated protein kinase (AMPK) and mammalian target of rapamycin complex 1 (mTORC1) pathway may be involved in this process. Recent studies using Extracellular Flux Analyzer demonstrated that metformin alters the activities of glycolysis, oxidative phosphorylation (OXPHOS), lipid oxidation, and glutaminolysis, which tightly link to the modulation of cytokine production in CD4+ and CD+8 T cells in various disease states, such as virus infection, autoimmune diseases, aging and cancers.

## 1. Introduction

The traditional herbal medicine in Europe, *Galega officinalis,* was found to be rich in guanidine and shown to lower blood glucose levels. Several guanidine derivatives were synthesized and used to treat diabetes in 1920s and 1930s, but they were discontinued due to severe toxicity [1]. The blood glucose lowering effects were recognized by the French physician Jean Sterne who reported the use of metformin to treat diabetes in 1957 [2]. However, metformin received the limited attention and the biguanides were generally discontinued in late 1970s due to the risk of lactic acidosis. The beneficial effects of metformin ameliorating insulin resistance without weight gain and hypoglycemia in the patients with type 2 diabetes were recognized in Europe and the long-term cardiovascular benefits [3] were identified by the UK Prospective Diabetes Study (UKPDS) in 1998, providing a new rationale to adopt metformin as an initial therapy to manage hyperglycemia in type 2 diabetes. Metformin is beneficial for aging-related morbidities such as obesity, metabolic syndrome, cardiovascular disease, and cognitive impairment [4] by its favorable action on the endothelial dysfunction [5]. In addition to risk reduction of any diabetes-related endpoint, myocardial infarction, and death from any cause in metformin [6], the reduction of cancer incidence was demonstrated in an observational study in Scotland [7] and a Chinese meta-analysis showed an overall reduction of 20% in cancer incidence in metformin users [8]. 

The metabolic effects of metformin with insulin sensitizing actions result in reduction of insulin and free insulin-like growth factor (IGF-1) levels may indirectly contribute the decreased incidence of cancer. In addition to the indirect effects, metformin as adenosine 5′-monophosphate-activated protein kinase (AMPK) activator and subsequent inhibition of mammalian target of rapamycin complex 1 (mTORC1) specifically retard the growth of malignant cells. Rapidly proliferating malignant cells prefer to facilitate the process of catabolic glucose metabolism. Metformin inhibits glycolysis rate limiting enzyme of glycolysis, hexokinase 2 (HK2), and mitochondrial respiratory complex 1, which result in the reduction of mitochondrial ATP production in cancer cells [9]. Recently, it has been demonstrated that metformin may also enhance the antitumor immune response by acting on the tumor infiltrating and circulating CD8 T cells. Furthermore, the advent of immune checkpoint inhibitors in cancer therapies and their prominent antitumor effects also arouse the interests to the immune mediated effects of metformin in various diseases. In this review article, we first discuss the molecular action mechanisms of metformin on energy metabolism and mitochondrial function in type 2 diabetes. Next, we further review the emerging roles of metformin, i.e., immune-mediated beneficial effects, in various diseases such as virus infection, autoimmune diseases, aging and cancers.

## 2. Molecular Mechanisms of Metformin-Induced Inhibition of Gluconeogenesis and Lipogenesis in Hepatocytes

Metformin is transported by facilitated diffusion via plasma membrane monoamine transporter (PMAT) and organic transporter 3 (OCT3) in enterocytes and further transported into hepatocytes via portal vein, where metformin reaches 40-70 μM, through OCT1 and OCT3 [10]. The excretion of metformin from hepatocytes to bile or circulation occurs through multidrug and toxin extrusion 1 (MATE1) and metformin concentration is reduced to 10-40 μM. Then, metformin enters renal epithelial cells via OCT2 and is secreted by renal MATE1 and MATE2 in unchanged form and eliminated into urine [10]. 

In hepatocyte, metformin partially inhibits mitochondrial respiratory-chain complex 1, resulting in reduction of ATP levels and accumulation of AMP and ADP. The gluconeogenesis is an energetically costly anabolic process and required 6 ATPs per molecule and depletion of ATP limits glucose synthesis. Pyruvate carboxylase (PC), Phosphoenol pyruvate carboxykinase (PEPCK) (Figure 1), Glyceraldehyde phosphate dehydrogenase require 2 ATPs, 2 GTPs, and 2 ATPs, respectively. The increased level of AMP inhibits an important rate-limiting gluconeogenic enzyme, Fructose-1, 6-bisphsphatase (FBPase). This gluconeogenic step antagonizes the opposite reaction that forms fructose-1,6-bisphosphate from fructose-6-phosphate and ATP in Phosphofructokinase (PFK)-dependent manner, a key rate-limiting step in glycolysis. Metformin was shown to reduce the glucagon signal transduction by decreasing 3′-5′-cyclic adenosine monophosphate (cAMP) thorough inhibiting adenylate cyclase coupled with glucagon receptor [11]. Decreased cAMP content leads to reduced activity of cAMP-dependent protein kinase A (PKA), an important signal transducer of glucagon-induced gluconeogenesis [12]. Metformin selectively inhibits the mitochondrial isoform of glycerophosphate dehydrogenase (mGPD) but not cytosolic GPD (cGPD), an enzyme that catalyzes the conversion of glyceraldehyde 3-phosphate (G3P) to dihydroxyacetone phosphate (DHAP). The inhibition of mGPD induces the accumulation of G3P, NADH and reduced conversion of lactate to pyruvate. Since DHAP and pyruvate are required for gluconeogenesis, it results in reduction of use of glycerol and lactate as gluconeogenic precursors [13]. In addition, metformin-induced increase in AMP/ATP ratio also activates AMPK, which suppresses lipid and protein synthesis and enhances glycolysis, fatty acid oxidation, mitochondrial biogenesis and autophagy [14]. The activation of AMPK by metformin resulted in inhibition of Acetyl-CoA carboxylase (ACC), and reduction of Malonyl-CoA. Since Malonyl-CoA inhibits the activity of carnitine palmitonyl transferase 1 (CPT1), CPT1 is activated by the reduction of Malonyl-CoA and fatty acid oxidation is enhanced. In addition, the activation of AMPK by metformin increased the expression of Sterol regulatory element binding protein-1c (SREBP-1c) which inhibits the expression of lipogenic and fatty acid synthesis genes (Figure 2).

Recently, the intestine has been focused as a target organ of metformin in addition to liver. Intraduodenal infusion of metformin activated the duodenal mucosal AMPK and lowered hepatic glucose production. Both glucagon-like peptide-1 receptor-PKA signaling and a neuronal-mediated gut-brain-liver pathway are required for metformin to lower hepatic glucose production [15]. The 3-day treatment with metformin in newly diagnosed patients with type 2 diabetes reduced *Bacteroides fragilis* and increased the bile acid glycoursodeoxycholic acid (GUDCA) in the gut, which were accompanied by inhibition of intestinal farnesoid X receptor (FXR) signaling [16]. By using Positron emission tomography-magnetic resonance imaging (PET-MRI), metformin increased the accumulation of [^18^F] fluorodeoxyglucose (FDG) in the intraluminal space of the intestine, suggesting that stool is one of the major disposal sites of glucose by the action of metformin targeting intestine [17].

## 3. Metformin and Mitochondrial Biogenesis and Dynamics

Various pharmacological activators of mitochondrial biogenesis such as AMPK activators, SIRT1 activators, nuclear receptor agonists, and cGMP modulators are possible candidate targets for the treatment of obesity, type 2 diabetes, and vascular complications [18]. Long-term administration of metformin demonstrated the increased activity of peroxisome proliferators-activated receptor γ (PPARγ) coactivator-1α (PGC-1α) and enhanced biogenesis of mitochondira [19]. in vitro models of vascular complications of diabetes demonstrated that metformin activates AMPK in human umbilical vein endothelial cells and reduces hyperglycemia-induced mitochondrial ROS production and mitochondrial biogenesis [20].

In addition to mitochondrial biogenesis, mitochondrial dynamics such as fusion and fission of mitochondria, and mitochondria-associated endoplasmic reticulum (ER) membranes (MAMs) are also involved in the process of diabetes. Once close contact between mitochondria is established, the dynamin-related outer mitochondrial membrane (OMM) proteins, such as mitofusin 1 (MFN1) and mitofusin 2 (MFN2), form homotypic (MFN1-MFN1 and MFN2-MFN2) or heterotypic (MFN1-MFN2) complexes. After tethering, inner mitochondrial membrane (IMM) fusion is mediated by optic atrophy 1 (OPA1) depending on the inner membrane potential [21]. The process of fusion retains the capacity of the mitochondria and maintains genetic and biochemical homogeneity by allowing the dilution of superoxide species and mutated DNA and the repolarization of membranes [22]. The reverse process of mitochondrial fusion, the division of mitochondria (fission) produces one or more daughter mitochondria, requires cytosolic dynamin-related protein 1 (DRP1). In the organellar interactions, MAMs function as membrane contact sites between the ER and mitochondria. The ER-mitochondria contact sites have emerged as major players in lipid metabolism, calcium signaling, autophagy and mitochondrial dynamics [23].

Disturbances in mitochondrial architecture and mitochondrial fusion-related genes are observed in situations of type 2 diabetes and obesity, leading to a highly fissioned mitochondria. Liver specific *Mfn1* knockout mice (Mfn1LKO) were associated with increased complex I abundance, sensitive to hypoglycemic effects of metformin, and protected against insulin resistance induced by a high-fat diet [24]. In cybrid cells harboring mitochondrial haplogroup B4, which are more likely to develop type 2 diabetes in Chinese population, presented increased mitochondrial fission profiles, while metformin inhibited mitochondrial fission and attenuated pro-inflammation profile [25]. In IR Huh7 cells with high invasiveness ability, mitochondrial fission was increased revealed by structured illumination microscopy and metformin could inhibit mitochondrial fission, which is the feature of type 2 diabetes [26]. The disruption of MAMs by pharmacological inhibition and genetic ablation of the mitochondrial MAM protein, cyclophilin D, causes the impairment of insulin signaling and metformin improves both MAM integrity and insulin sensitivity [27].

## 4. Metformin and Metabolism in Cancer Cells

OCT1 is almost exclusively expressed in the liver, and it is a major target tissue for metformin. In contrast to the virtual absence of OCT1 in various tissues except liver, the tumor cells significantly express OCT1 and it may be related the antitumor effects of metformin [28]. OCT1 is also involved in the uptake of irinotecan and paclitaxel and OCT1-positive cancer cells exhibit significantly higher susceptibilities to the cytotoxic effects of these anticancer agents [29]. Liver kinase B1 (LKB1) is identified as a tumor suppressor gene and upstream activator of AMPK. The activation of LKB1-AMPK pathway by metformin results in the inhibition of Raptor-mTORC1 complex and suppression of cellular protein synthesis and cell growth. Phosphorylation of AMPK activates tuberous sclerosis complex 2 (TSC2) in a subunit of the TSC1–TSC2 complex, which further inactivates the small GTP-binding protein Ras Homolog Enriched in Brain (RHEB). The inactivated RHEB fails to promote the activity of mTORC1, suppresses the cell cycle, and reduces the proliferation of the cancer cells [9]. In addition to identification of tumor suppressor genes in cancer development, metabolic alterations induced by cancer cells was also rediscovered. The observation by Otto Warburg demonstrated that the proliferating cancer cells highly consume glucose and produce plenty of lactate and it is the reverse of Pasteur effects, where the fermentation is inhibited in the presence of O_2_ [30]. Recent investigations demonstrated that tumor suppressors and oncogenes converge on the prolyl hydroxylases and hypoxia-inducible factor (HIF), reverse the Pasteur effects, and thereby induce the Warburg effects. Importantly, the cancer cells carry out and enhance both aerobic glycolysis and mitochondrial respiration concurrently [30]. The tumor cells utilize the glucose for the proliferation and hypertrophy of the cells. The phosphorylation of glucose, the production of glucose-6-phosphate (G6P), is the initial step in glucose metabolism in cancer cells [9]. Hexokinase 1 (HK1) and HK2 are responsible for the production of G6P and show high affinity for glucose with Km values of μM range. They are highly sensitive to inhibition by their own product, G6P. HK1 is ubiquitously expressed, while HK2 is detected in skeletal muscle, adipose tissue and heart. HKs 1 and 2 are found in mitochondrial fraction and interact with the permeability transition pore including the voltage-dependent anion channel 1 (VDAC1) responsible for ATP flux to the cytoplasm from mitochondria. Such association between HK2 and VDAC1 facilitates the production of G6P by HK2 and protects cells from apoptosis. HK2 is highly expressed in lung and breast cancer and required for the proliferating cancer cells. Metformin directly inhibits HK2 activity by occupying the G6P binding site and induces dissociation of HK2 from mitochondria [9]. In addition to the reduction of glycolysis, metformin inhibits mTORC1 by decreasing the insulin and IGF-1 concentrations and inhibiting IGF1-induced AKT phosphorylation. Since AKT further phosphorylates HK2 at Thr473 and facilitates the association of HK2 with mitochondria, metformin decreases HK2 expression, activity and mitochondrial interaction [31]. Finally, metformin inhibits mitochondrial ATP production acting on respiratory chain complex 1 and decreasing TCA cycle intermediates, thus, metformin inhibits both glycolysis and mitochondrial respiration in cancer cells.

## 5. Glycolysis and Cytokine Production in CD4 and CD8 T Cells

The metabolic effects of metformin in type 2 diabetes targeting hepatocytes and cancer cells are discussed in previous sections. Next, the link between metabolic alterations and functional changes induced by metformin in CD4 and CD8 T cells is discussed. The quiescent lymphocytes such as naïve and memory CD4 T cells store ATP reserves via oxidative phosphorylation (OXPHOS) and fatty acid oxidation (FAO) to prepare for activation. Like cancer cells, once activated lymphocytes escape from quiescent state, proliferate, produce various cytokines, and shift to aerobic glycolysis. Under the state of low glycolytic flux in resting CD4+ cells, G3P dehydrogenase (GAPDH) associates with the 3′ untranslated region (UTR) of IFNγ mRNA and prevents its translation. Upon activation of CD4+ cells, the signaling through T cell receptor stimulates aerobic glycolysis, G3P provided by glycolysis binds to GAPDH as a substrate and inhibits the binding of GAPDH to IFNγ mRNA [32]. The glycolytic activities also regulate the differentiation and effector function of CD8+ T cells. Following the antigen stimulation, naïve CD8+ T cells rapidly increase uptake of glucose and glutamine and provide ATP and fatty acid, in which metabolic reprograming supports the cell proliferation, which is characterized with short survival and reduced antitumor activities. As effector T cell response subsided, memory T cells increase mitochondrial integrity and metabolism, and activate OXPHOS and FAO to sustain prolonged cell survival and antitumor activities.

Under a glucose-poor tumor microenvironment, the aerobic glycolysis is reduced, and tumor-infiltrating T cells are associated with limited tumoricidal effector function. The glycolytic metabolite phosphoenolpyruvate (PEP) inhibits sarco/ER Ca^2+^-ATPase (SERCA) activity, reduces Ca^2+^ influx into ER. The increased cytosolic Ca^2+^ activates nuclear factor of activated T cells (NFAT) and it provokes antitumor responses [33]. The overexpression of phosphoenolpyruvate carboxykinase 1 (PCK1)-overexpressing T cells restricted tumor growth and prolonged the survival of melanoma-bearing mice. Metformin at around 10 μM concentration achieved by oral administration enhanced production of IFNγ, TNFα and expression of CD25 of CD8+ T cells upon TCR stimulation [34].

## 6. Metformin and Immune-Mediated Benefits in Various Diseases

Metformin has been a long-standing and fist-line drug for treatment of type 2 diabetes and its beneficial effects on virus infection, autoimmune diseases, aging and cancers are also recognized. The underlying mechanism especially in anti-cancer effects was not fully explored. Since metformin was found to enable the normal but not T-cell-deficient SCID mice to reject solid tumors, the beneficial effects seem to be immune-mediated [35]. A direct action of metformin on CD8+ tumor infiltrating lymphocytes is critical for protection against functional exhaustion and recovery of multiple cytokine production. Since the discovery, vigorous efforts have been made to elucidate the action mechanism of metformin on the differentiation and activation of CD4+ and CD8+ T cell population in various diseases. In various autoimmune diseases, the effects of metformin on CD4+ Th1, Th17 and regulatory T cells (Tregs) have been focused. Adenosine 5′-monophosphate-activated protein kinase (AMPK) is a master sensor of cellular energy states and activated by reduced AMP/ATP, a shortage of cellular nutrients. The activation of AMPK restores energy balance by inhibiting anabolism and enhancing catabolism to produce energy. The activation of AMPK also interferes T cell activation and differentiation by inhibiting mammalian target of rapamycin (mTOR) and subsequently inhibiting glycolysis and enhancing lipid oxidation [36,37]. In autoimmune insulitis model of NOD mice, metformin inhibits the differentiation of Th1 and Th17 cells while promotes the development of Tregs through the activation of AMPK [38]. In contrast, CD8+ cells are focused on tumor infiltrating lymphocytes (TILs) in various malignancies. The metformin-induced production of multiple cytokines in CD8+ TILs may not be totally dependent on the AMPK activation and dependent on the nutrient condition in cancer cells, which actively engulf glucose in tumor environment. AMPK activation requires more than 1 mM at unachievable high concentration of metformin in vivo, while metformin at around 10 μM, achievable concentration in vivo, enhanced production of interferon (IFN) γ, tumor necrosis factor (TNF) α and expression of CD25 of CD8+ T cells upon TCR stimulation [34]. Under a glucose-rich condition, glycolysis was exclusively involved in enhancing IFNγ production in CD8+ TILs; however, under a low-glucose condition, fatty acid oxidation or autophagy-dependent glutaminolysis were also involved [34]. In following sections, we describe how metformin regulates the interface of metabolism and immune function in the CD4 and CD8 T cells under the various diseases condition, such as virus infection, autoimmune diseases, aging and cancers.

## 7. HIV (Human Immunodeficiency Virus) Infection and Metformin

Metformin was first recognized by its metabolic benefits in the HIV-infected patients receiving HAART (highly active anti-retroviral therapy) including protease inhibitors [39]. HIV patients treated with protease inhibitors were characterized as treatment-indcued metabolic abnormalities such as prominent insulin resistance, dyslipidemia and lipodystrophy associated with accumulated visceral fat and reduced subdermal fat tissues. The administration of metformin was not associated with significant changes in visceral or subcutaneous abdominal fat, but oral glucose tolerance test demonstrated the tendency for the reduction of insulin area under the curve (AUC) [40]. In prediabetic patients with HIV, the administration of metformin resulted in improvement of HbA1c and insulin resistance and it may prevent the progression from prediabetes to diabetes [41]. In the non-diabetic patients with HIV, the administration of metformin resulted in amelioration of HAART-induced weight gain associated with increased abundance of anti-inflammatory bacteria such as butyrate-producing species and the protective *Akkermansia muciniphila* [42]. In addition to metabolic benefits, metformin may have benefits on the HIV reservoir in long-lived CD4+ T cells with an increased risk of inflammation-associated complications and non-AIDS no-morbidities, since metformin modulates T-cell activation by regulating intracellular immunometabolic checkpoints such as the AMPK and mTOR (mammalian target of rapamycin), in association with microbiota modification [43]. The administration of metformin in 12 virally suppressed HIV-infected individuals reduced the frequency of PD1 (programmed cell death protein 1)+, PD1+TIGIT+ (T cell immunoreceptor with Ig and ITIM domains), and PD1+TIGIT+TIM3+ (T cell mucin-domain containing-3) expressing CD4 T cells, suggesting the amelioration of CD4 T cell exhaustion [44] (Table 1). In LILAC pilot study, metformin preferentially acts on the intestine and decreases mTOR activation selectively occurs in colon infiltrating CCR6+CD4+ Th17 T cells [45]. The metformin therapy over the 8-week course in 7 euglycemic, virally suppressed HIV-infected participants on HAART significantly improved ex vivo polyfunctional HIV-Gag-specific CD8 T cell responses to anti-PD-L1 mAb, suggesting the enhancement of anti-HIV CD8 T cell immunity [46]. In addition, the elevated OXPHOS pathway is associated with poor outcomes in the patients with HIV and the administration of metformin targeting OXPHOS suppresses HIV-1 replication in human CD4+ T cells and humanized mice [47].

In addition to HIV infection, it has been reported that metformin reduced the risk of mortality rate in acute respiratory distress syndrome (ARDS) in corona virus disease (COVID-19). Metformin blocked LPS-induced and ATP-dependent mitochondrial DNA synthesis and generation of oxidized mtDNA, an NLR family pyrin domain-containing protein 3 (NLRP3) ligand by targeting electron transport chain complex 1 [48].

**Table 1 ijms-24-00755-t001:** Effects of metformin in infection and immune-mediated diseases.

Diseases	Species	Targeted Immune-Mediated Cells	Possible Beneficial Effects
Human immunodeficiency virus infection (HIV)	Mouse	-	-
	Human	Peripheral PD-1+CD4+↓, PD-1+TIGIT+CD4+↓, PD-1+TIGIT+TIM3+CD4+↓	Reduction of CD4 T cell exhaustion in HIV patients [44]
Rheumatoid arthritis (RA)	Mouse	RORγt+IL-17+CD4+↓	Reduction of Th17 differentiation and attenuation of arthritis scores and bone destruction collagen antibody-induced arthritis (CAIA) mouse model [49]
	Human	Osteoclastogenesis in CD14+ monocytes treated with macrophage colony stimulating factor (M-CSF) and soluble receptor activator of NF-κB ligand (sRANKL)↓, tumor necrosis factor-α (TNF-α)-induced expression of inflammatory cytokines from human fibroblast-like synoviocyte MH7A cells↓	Reduction of joint inflammation and destruction in human cell culture [50]
Systemic lupus erythematosus (SLE)	Mouse	Tf↓, Th17↓, Treg↑, B cell differentiation into plasma cells and germinal centers formation↓	Suppression of systemic autoimmunity in *Roquin^san/san^* mice [51]
	Human	Neutrophil extracellular traps (NETs) mtDNA release from neutrophils↓, IFN-α production from plasmacytoid dendritic cells (PDCs)↓	Down-regulation of the NET mtDNA–PDC–IFNα pathway in SLE patients [52]
Autoimmune insulitis	Mouse	Th1↓, Th17↓, Treg↑	Mitigated autoimmune insulitis in non-obese diabetic (NOD) mice [38]
	Human	-	-
Inflammatory bowel disease (IBD)	Mouse	Interferon (IFN)-γ production from mucosal CD4+↓	Amelioration of T cell-transfer model of chronic colitis in severe combined immunodeficient (SCID) mice [53]
	Human	TNF-α↓, transforming growth factor β1 (TGF-β1)↓, malondialdehyde (MDA)↓, myeloperoxidase (MPO)↓ in colon biopsy samples	Improvement of histopathology in female patients suffering from ulcerative colitis by indole-3-carbinol and/or Metformin [54]
Sjögren’s syndrome (SS)	Mouse	Th1↓, Th17↓, Treg↑	Amelioration of salivary gland inflammation and hypofunction in NOD/ShiLtJ mice, an animal model of SS [55]
	Human	-	Reduced risk of SS in type 2 diabetic patients in population-based cohort study [56]
Allergic airway inflammation	Mouse	Treg↑	Improvement of asthma airway inflammation in obese asthmatic mouse model [57]
	Human	-	-
Hashimoto’s thyroiditis (HT)	Mouse	Th17↓, M1↓	Amelioration of HT in mouse model by high-iodine water feeding and thyroglobulin immuno-injection [58]
	Human	-	-
Scleroderma	Mouse	Treg↑, Teff↓	Anti-fibrotic effects in bleomycin-induced scleroderma mouse model [59]
	Human	-	-
Multiple sclerosis (MS)	Mouse	Th17↓, Treg↑	Protective effects in mouse experimental autoimmune encephalomyelitis (EAE) model [60]
	Human	Myelin basic protein (MBP) peptide-specific cells secreting IFN-γ and IL-17↓, Treg↑	Anti-inflammatory effects in patients with MS [61]
Acute graft-versus-host diseases (aGVHD)	Mouse	Th1↓, Th17↓, Treg↑, Th2↑	Attenuation of aGVHD in allo-bone marrow transplantation mouse model [62]
	Human	Th17↓, Treg↑	Improvement of immunological balance by increasing Treg cells and decreasing Th17 cells in liver transplantation patients [51]
Allergic airway inflammation	Mouse	Treg↑	Alleviation of airway inflammation in obese asthmatic mouse model by administering a high-fat diet (HFD) and ovalbumin (OVA) sensitization [57]
	Human	-	-
Ovarian fibrosis	Mouse	B cells↑, T cells↑, metformin-responsive macrophage III subpopulation↑	Amelioration of ovarian fibrosis in aged mouse model [63]
	Human	CD206+:CD68+ cell ratio↓, CD8+ infiltration↓	Abrogation of age-associated ovarian fibrosis in postmenopausal human ovaries [64]

T cell immunoreceptor with Ig and ITIM domains (TIGIT); T cell mucin-domain containing-3 (TIM3).

## 8. Autoimmune Diseases and Metformin

The data for the efficacy of metformin in autoimmune diseases are limited and most of them were performed in animal studies (Table 1). In the collagen antibody-induced arthritis (CAIA) murine model, metformin attenuated arthritis scores, bone destruction, serum levels of the pro-inflammatory cytokines, such as TNF-α and IL-1, and the number of RORγt+CD4+ T cells from axillary lymph nodes. Metformin treatment of splenocytes cultured in Th17-differentiation-inducing conditions decreased the number of RORγt+CD4+ T cells in a dose-dependent manner and downregulated STAT3 phosphorylation via the AMPK pathway [49]. In CD4+ T cells from the patients with systemic lupus erythematodes (SLE), metformin at 4 mM inhibits the transcription of IFN-stimulated genes independent of AMPK activation and mammalian target of mTORC1 inhibition and it was replicated by inhibitors for electron transport chain respiratory complexes I, III and IV, suggesting the importance of OXPHOS in the production of IFN signature genes [65]. In addition, 2-deoxy-D-glucose (2-DG), glycolysis inhibitor, also prevent the activation of CD4+ T cells and reduced both IFN-γ and IL-17 production in B6.Sle1Sle2.Sle3 mice [66,67]. The combination of 5 mM metformin and 2-DG treatment more potently suppressed IFN-γ production and cell proliferation in activated primary human CD4+ T cells, suggesting multiple metabolic networks of activated in human T cells may be the target for the treatment of SLE [68]. Since CD28 costimulation through phosphatidylinositol 3’-kinase (PI3K) and Akt is required for T cells to increase glucose uptake and glycolysis, the combination of CTLA4Ig, inhibitor for CD28 signaling, and metformin decreased the development of lupus nephritis in (NZB x NZW)F1 mice treated at the early stage of disease [69]. The amelioration of autoimmune and allergic animal models by metformin was reported in T cell-transfer model of chronic colitis in SCID mice injected with CD4+ CD45RB(high) T cells [53], non-obese diabetic (NOD)/ShiLtJ mice (an animal model of Sjogren’s syndrome) [55], autoimmune insulitis in NOD mice^38^, concanavalin A (Con A)-induced hepatitis, an experimental model of T cell-mediated liver injury, in C57Bl/6 mice [70], and obese asthmatic mouse model by administering a high-fat diet (HFD) and ovalbumin (OVA) sensitization [57]. In a mouse Hashimoto’s thyroiditis model induced by high-iodine water feeding and thyroglobulin immuno-injection, metformin reduced thyroglobulin antibody production and lymphocyte infiltration in thyroid associated with reduced number and function of Th17 cells and M1 macrophages polarization [58].

## 9. Aging and Metformin

In the process of aging, the inflammation plays fundamental roles in all the age-associated diseases (Table 1). Cytokine profiling demonstrated that CD4+ T cells derived from older subjects mimic a diabetes-associated Th17 profile such as IL-6, IL-17A, IL-17F, IL-21, and IL-23 [71]. Although inflammation is traditionally fueled by glycolysis, in CD4+ T cells from older subjects, higher OXPHOS and lower glycolysis was observed. Furthermore, metformin at 100 μM shifts older CD4+ T cells to younger CD4+ T cells with lower OXPHOS and higher glycolysis associated with amelioration of Th17 profile [71]. The amelioration of the Th17 inflammaging profile by metformin is mediated by increasing autophagy and improving mitochondrial bioenergetics [71,72]. In aging process, the reduction of brown adipose tissue links to the development of obesity and diabetes. Metformin inhibits an inflammatory program executed by hypoxia-inducible factor-1α (HIF1α) in M1-polarized macrophages and exerts beneficial effect on insulin-mediated glucose uptake and β-adrenergic responses in brown adipocytes [73]. In the process of aging, development of ovarian fibrosis is the risk for the ovarian cancer in the postmenopausal women. Metformin abrogates age-associated ovarian fibrosis in aged C57/lcrfa mice demonstrating reduced CD8+ T-cell infiltration and reduced CD206+:CD68+ cell ratio [64]. In the cohort of normal human ovaries, ovarian fibrosis developed associated with increased CD206+:CD68+ cell ratio, and CD8+ T-cell infiltration, which were reduced in the patients with metformin use [64]. Metformin may be useful for the aging-dependent ovarian cancer prophylaxis. Aging also impacts the alloimmunity and the activation of CD4+ T cells from old mice but not young CD4+ T cells exclusively relied on glutaminolysis. DON (6-diazo-5-oxo-l-norleucine), a glutaminolysis inhibitor, resulted in reduced IFN-γ production and prolonged graft survival, while both inhibition of glycolysis (2-DG) and OXPHOS (metformin) in combination with DON was required for the elongation of graft survival in young animals [74].

## 10. Immune-Mediated Antitumor Effects of Metformin

Metformin, a long-standing prescribed drug for T2D, had been reported to have anti-cancer effects by epidemiology studies. Metformin was reported to demonstrate an ability to reject various solid tumors in normal, but not T-cell-deficient SCID mice and it increased the numbers of CD8+ tumor-infiltrating lymphocytes (TILs), with multiple cytokine production, i.e., IL-2, TNFα, and IFN-γ [35]. In high fat-high glucose (HFS) induced C57BL/6JJcl obesity mice, multifunctionality of CD8+ splenic and TILs was impaired and associated with enhanced tumor growth. In CD8+ splenic T cells from the HFS mice, glycolysis/basal respiration ratio was significantly reduced, and glycolysis were reversed by metformin [75] (Figure 3). In contrast to CD8+ T cells, metformin inhibited tumor growth, decreased IL-22 production and de novo generation of Th1- and Th17 cells from naïve CD4+ cells in a mouse hepatocellular carcinoma model [76]. Furthermore, metformin inhibits the differentiation of naive CD4+ T cells into inducible Treg (iTreg) by reducing forkhead box P3 (Foxp3) protein, caused by mTORC1 activation [77]. In human studies of TILs, metformin triggered reduced CD8+ effector T cells and FoxP3+ Tregs in head and neck squamous cell carcinoma [78], CD3+ CD8+ TILs in colorectal cancer in patients with T2D [79], CD8+ TILs and tumor-suppressive CD11c+ macrophages in human esophageal cancer [80], and memory stem and central memory CD8+ T cells [81]. In addition to CD4+ and CD8+ T cells, metformin increased natural killer (NK) cells and their cytotoxicity in the patients with head and neck cancer squamous cell carcinoma (HNSCC) [82]. Low-dose of metformin also increased tumor-suppressive (CD11c+) and a decreased tumor-promoting (CD163+) macrophages in the patients with esophageal squamous cell carcinoma (ESCC) [80]. Mannose-modified macrophage-derived microparticles (Man-MPs) loaded metformin (Met@Man-MPs) efficiently repolarized M2-like tumor-associated macrophages (TAMs) to Mi1-like phenotype [83].

The combination of metformin with anticancer therapies and immunotherapy has been vigorously attempted. The combination therapy of cisplatin (CDDP) and metformin, cisplatin/polystyrene-polymetformin (HA-CDDP/PMet) dual-prodrug co-assembled nanoparticles, in Lewis lung carcinoma (LLC) injected C57/BL6 mice, resulted in tumor cell apoptosis associated with increased CD4+ and CD8+ T cells, a concomitant decrease in Tregs with enhanced expression of the cytokines IFN-γ and TNF-α [84]. In contrast, the combination therapy of melatonin, metformin, and dacarbazine in disseminated melanoma patients, no benefit was observed over dacarbazine monotherapy; however, the increase of CD3+ CD4+ HLA-DR+, CD3+ CD8+ HLA-DR+, CD3+ CD8+, CD4+ CD25high CD127low was observed in patients with clinical benefit [85]. The combination of local radiation and metformin in LuM1, a highly lung metastatic subclone of colon 26, injected BALB/c mice also resulted in delay in tumor growth associated with enhanced IFN-γ production of the splenic CD4+ and CD8+ T cells [86]. Melittin is the main component and the major pain producing substance of honeybee (*Apis mellifera*) venom and *Clostridium novyi*-spores coated with melittin-RADA32 nanofiber hybrid peptide and metformin (MRM) were applied to C57BL/6 mice injected with GL261 cells, glioblastoma cell line. The antitumor effects of MRM-coated spored were mediated by inducing sustainable CD8+ T cell responses and promoting M1 macrophage polarization in glioblastoma models [87].

The combination of metformin and the blockade of programmed death-1 (PD-1) and programmed death-ligand 1 (PD-L1) is promising strategy for the cancer treatment. In BALB/c mice injected with CT26 (murine colon cancer cell line), the combination of 3-hydroxy-butyrate (3-OBA) blocking lactate/GPR81 and metformin was found to reduce the production of lactic acid and recovered the inhibitory effect of metformin on PD-1 expression. Metformin changes the expression pattern of immune mediators in hepatocellular carcinoma (HCC) immune microenvironment, including PD-1, cytotoxic T lymphocyte antigen-4 (CTLA-4) [88]. Furthermore, metformin reduced the expression of PD-L1 in cancer cells by disrupting the electrostatic interaction and enhancing membrane dissociation of cytoplasmic domain [89]. Metformin also activated AMPK, which directly phosphorylates S195 of PD-L1 and results in the endoplasmic reticulum (ER) accumulation and ER-associated protein degradation (ERAD) in cancer cells [90]. The combination of metformin and cisplatin-based chemotherapy^81^, chitosan oligosaccharide [91,92], photodynamic immunotherapy [93,94], vaccine immunotherapy [95], and 2-deoxy-D-glucose (2DG) [96] further enhanced the reduction of PD-L1 expression in cancer cells. The dual administration of P PD-1/PD-L1 blocking peptide C8 and 3-OBA further potentiated theantitumor activity of metformin [97]. The mannose-modified macrophage-derived microparticles (Man-MPs) loading metformin (Met@Man-MPs) targeted M2-like TAMs to repolarize into M1-like phenotype, increase the recruitment of CD8+ T cells into tumor tissues, and boost anti-PD-1 antibody therapy in H22 tumor-bearing BALB/c mice [83].

## 11. Conclusions

Metformin demonstrates prominent impacts on the differentiation and activation of CD4+ and CD+8 T cells in various disease states, such as virus infection, autoimmune diseases, aging, and cancers. The increased glycolysis and reduced oxidative phosphorylation (OXPHOS) by metformin links to the activation of CD4+ and CD+8 T cells and enhanced cytokine production. The combination of metformin with anti-viral drugs, immunosuppressants, anticancer agents and immune checkpoint inhibitors enhances the beneficial effects in the treatment of these diseases. Metformin also targets other immune-mediated cells, such as macrophages, and somatic cell types including hepatocytes, Intestinal epithelial cells, and adipocytes. The organellar function of mitochondria and ER is altered by metformin acting on the diverse molecular targets such as AMPK, mTORC1, and mGPD. The understanding of multiple actions of metformin on immune-mediated cells and somatic cells would further enhance the identification of new molecular targets and development of new therapeutic strategies for various diseases.

## Figures and Tables

**Figure 1 ijms-24-00755-f001:**
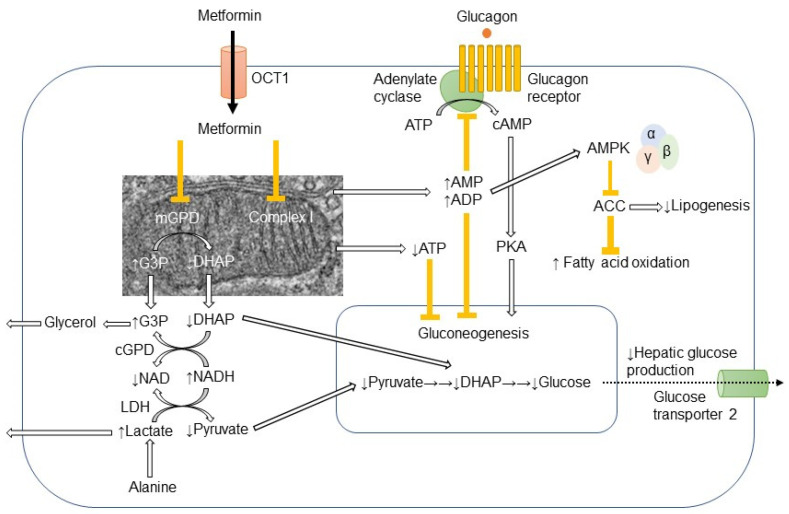
Action mechanism of metformin in hepatocyte.Metformin is mainly transported into hepatocyte through organic transporter 1 (OCT1) and inhibits respiratory-chain complex 1 (Complex 1). The reduction of ATP levels and increase in AMP and ADP levels result in the inhibition of gluconeogenesis. The increased AMP levels inhibit the adenylate cyclase coupled with glucagon receptor and the subsequent cAMP/protein kinase A (PKA) signaling pathway, which ultimately links to enhancement of gluconeogenesis. Metformin-induced elevation of AMP/ATP ratio also activates AMP-activated protein kinase (AMPK). Lipogenesis is inhibited and fatty acid oxidation is enhanced by the AMPK-induced inhibition of acetyl CoA carboxylase (ACC). Metformin also inhibits mitochondrial glycerophosphate dehydrogenase (mGPD) but not cytosolic GPD (cGPD). Glyceraldehyde 3-phosphate (G3P), nicotinamide adenine dinucleotide (NADH), and lactate are accumulated, while dihydroxyacetone phosphate (DHAP), NAD, and pyruvate are reduced. By the reduction of substrates for gluconeogenesis, i.e., pyruvate and DHAP, hepatic glucose production is reduced by metformin. The excess of glycerol and lactate enters into circulation.

**Figure 2 ijms-24-00755-f002:**
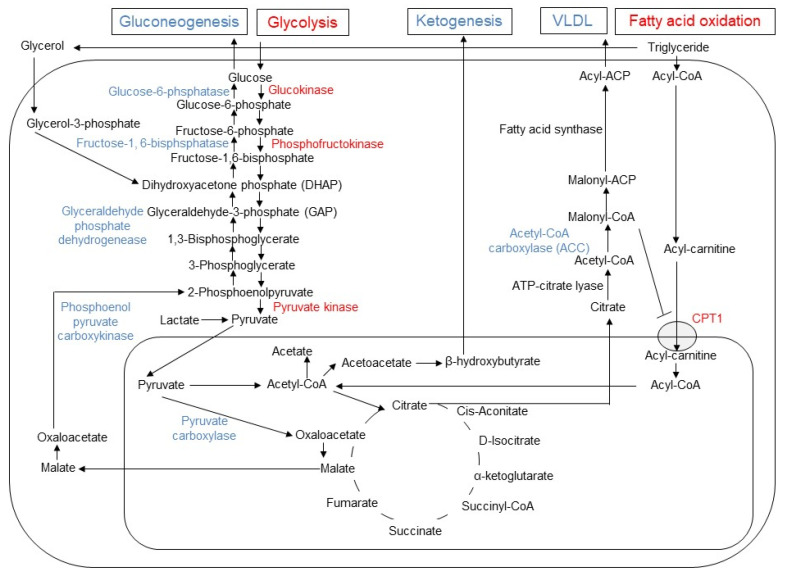
Changes in gluconeogenesis, glycolysis, lipogenesis, fatty acid oxidation, and ketogenesis induced by metformin in hepatocyte. In hepatocytes, metformin enhances glycolysis and fatty acid oxidation, while it inhibits gluconeogenesis, ketogenesis and release of very low-density lipoprotein (VLDL). The enzymes with enhanced activities are indicated by blue, while the reduced enzymes by red font. Since malonyl-CoA inhibits the activity of carnitine palmitoyltransferase 1 (CPT1), the reduced activity of acetyl CoA carboxylase (ACC) results in increased influx of acyl-carnitine into mitochondria and fatty acid oxidation by metformin.

**Figure 3 ijms-24-00755-f003:**
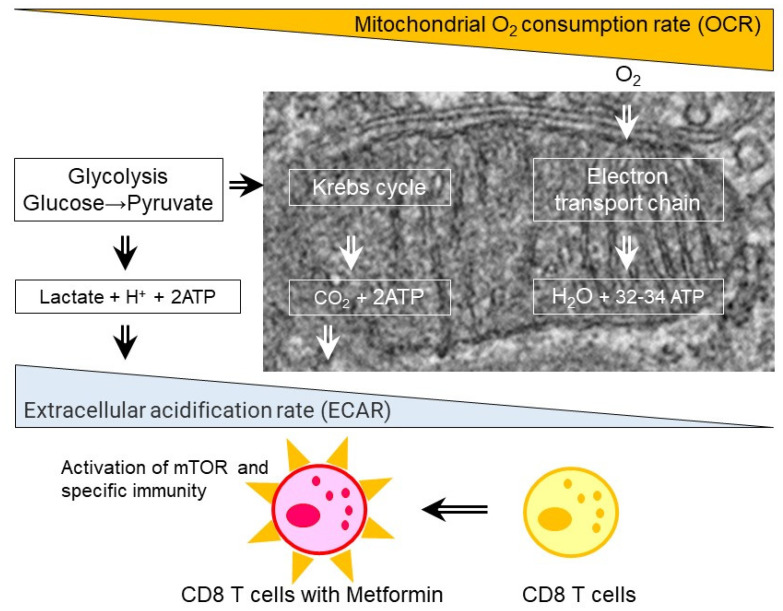
Effect of metformin on CD8 T cells. Metformin treatments in the patients with type 2 diabetes and obese mice fed with high fat diet results in increase in extracellular acidification rate (ECAR) and oxygen consumption rate (OCR) ratio, activation of mammalian target of rapamycin (mTOR), and enhanced specific immunity.

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
