# Peer review of "Metformin and Its Immune-Mediated Effects in Various Diseases"

_ijms, 2023, doi:10.3390/ijms24010755_

Round 1

Reviewer 1 Report

I enjoyed reviewing this manuscript. The article is interesting and well written. The topic is very hot as metformin is proving that its extra glycemic effects are no less promising than strictly metabolic ones. The sections are well structured. The figures are clear.

I only suggest inserting a couple of particularly up-to-date references in the text:

Diabetes Research and Clinical Practice Volume 160 February 2020 Article number 108025 DOI 10.1016/j.diabres.2020.108025

Biomedicines Volume 9, Issue 1, Pages 1 – 26 December 2021 Article number 3 DOI 10.3390/biomedicines9010003

Reviewer 2 Report

In this research, the authors reviewed the “Metformin and its immune-mediated effects in various diseases” In my opinion, the current stage of this paper could meet the requirements of International Journal of Molecular Sciences after major revisions.

My comments are as details:

1.      As newly proved, metformin and its derivatives could decrease PD-L1 expression in tumor cells. But, this important part was not discussed in this review. Some related researches may be helpful to the authors and some references related to these functions of Metformin should be added: 10.1038/S41467-021-25416-7; 10.1016/j.ijbiomac.2022.10.167; 10.1021/acsami.0c21743; 10.1016/j.molcel.2018.07.030; 10.1002/adfm.202007149; 10.1016/j.carbpol.2021.118869; 10.1136/jitc-2021-002614; 10.1016/j.jconrel.2022.11.004; 10.1016/j.intimp.2022.108889; 10.3390/cancers14051343.

2.      How Metformin affect the function of NK cells, DC cells, and macrophage should be clearly discussed in this review. Some related research was listed as below: 10.1136/jitc-2022-005632; 10.1158/1078-0432.CCR-20-0113; 10.1038/s41467-020-20723-x;

3.      How Metformin affect some other disease should be discussed. Some related publications are as bellows: 10.1016/j.redox.2021.102171; 10.1016/j.immuni.2021.05.004; 10.2139/ssrn.3782444;  10.1158/1078-0432.CCR-19-0603;

4.      The conclusion part was too plain. An in-depth outlook or conclusion should be added. 

5.      It’s better to add a table that clearly summarize the Metformin and its immune-mediated effects in various diseases diseases, especially the immune cells that related to its functions.
